# Statistical Copolymers of *N*–Vinylpyrrolidone and 2–Chloroethyl Vinyl Ether via Radical RAFT Polymerization: Monomer Reactivity Ratios, Thermal Properties, and Kinetics of Thermal Decomposition of the Statistical Copolymers

**DOI:** 10.3390/polym15081970

**Published:** 2023-04-21

**Authors:** Nikolaos V. Plachouras, Marinos Pitsikalis

**Affiliations:** Industrial Chemistry Laboratory, Department of Chemistry, National and Kapodistrian University of Athens, Panepistimiopolis Zografou, 15771 Athens, Greece; plachouras.v.nikolaos@gmail.com

**Keywords:** radical polymerization, RAFT, NVP, CEVE, reactivity ratios, thermal analysis, kinetics of thermal decomposition

## Abstract

The radical statistical copolymerization of *N*–vinyl pyrrolidone (NVP) and 2–chloroethyl vinyl ether (CEVE) was conducted using the Reversible Addition–Fragmentation chain Transfer (RAFT) polymerization technique, employing [(O–ethylxanthyl)methyl]benzene (CTA-1) and O–ethyl S–(phthalimidylmethyl) xanthate (CTA-2) as the Chain Transfer Agents (CTAs), leading to P(NVP–stat–CEVE) products. After optimizing copolymerization conditions, monomer reactivity ratios were estimated using various linear graphical methods, as well as the COPOINT program, which was applied in the framework of the terminal model. Structural parameters of the copolymers were obtained by calculating the dyad sequence fractions and the monomers’ mean sequence lengths. Thermal properties of the copolymers were studied by Differential Scanning Calorimetry (DSC) and kinetics of their thermal degradation by Thermogravimetric Analysis (TGA) and Differential Thermogravimetry (DTG), applying the isoconversional methodologies of Ozawa–Flynn–Wall (OFW) and Kissinger–Akahira–Sunose (KAS).

## 1. Introduction

Poly(*N*–vinyl pyrrolidone) (PNVP) is a valuable material of academic interest and with industrial applications. It is a biocompatible and hemocompatible polymer and is non-toxic and widely employed in the biomedical and pharmaceutical sectors [1,2]. It is synthesized exclusively via radical polymerization [3]. Among the various controlled radical polymerization techniques, the Reversible Addition–Fragmentation chain Transfer (RAFT) methodology is the best approach to control the molecular characteristics and synthesize complex macromolecular architectures [4,5,6,7,8]. Due to its solubility in water, the copolymerization with hydrophobic monomers can lead to the synthesis of amphiphilic materials. In fact, the synthesis of statistical copolymers based on NVP and various methacrylates has been reported in the literature, with the hydrophobic character predominating in each case due to the higher reactivity of the methacrylates [9,10,11,12,13]. Other more complex amphiphilic or double hydrophilic architectures have appeared in the literature [14].

Poly(vinyl ethers) (PVEs) constitute another class of monomers with significant practical applications. VE oligomers have been recommended as skin adhesives, since they lack intrinsic toxicity [15]. Particularly, cytotoxicity studies for poly(2–chloroethyl vinyl ether) (PCEVE) networks demonstrated that they comply with specifications for a short contact period with blood [16]. PCEVE is a hydrophobic, elastomeric compound which is capable of serving as an intermediate for the synthesis of more complicated structures, because the chlorine groups can be easily substituted by other groups under suitable experimental conditions [17]. Being a VE, CEVE has only been polymerized via cationic polymerization using a variety of systems and methods [17,18,19,20].

The combination of the RAFT technique with cationic polymerization [21] opened new horizons for the synthesis of well-defined novel polymeric products through the incorporation of monomers within the same structure that are susceptible to radical and cationic polymerization [22,23,24]. Modification of the RAFT polymerization procedure allows for the formation of a wide range of copolymers that would otherwise be impossible to synthesize in a single polymerization step [25]. The employment of double RAFT agents renders the simultaneous copolymerization of exclusively radically polymerizable with cationically polymerizable monomers possible [26]. However, this approach is limited to the synthesis of block copolymers [27,28]. The synthesis of statistical copolymers is not feasible through this methodology.

Radical polymerization of VEs has been considered unachievable due to the instability of the generated radicals and the existence of several side reactions, such as β-scission [29,30]. The relatively controlled radical polymerization has been reported only for hydroxy-functional VEs in aqueous media due to the formation of hydrogen bonds between the oxygen atom of the VE and the pendant hydroxyl group that reduced the reactivity of the growing radical, thus suppressing unfavorable side reactions, such as β-scission [31,32]. To achieve the radical polymerization of VEs without a hydroxyl group in the ether group, additives must be used, such as LiOH [33]. In this study, NVP and CEVE were copolymerized in bulk via Radical RAFT without the presence of additives, leading to the formation of biocompatible amphiphilic statistical copolymers. The optimal copolymerization conditions were explored in order to obtain the best molecular characteristics and the highest degree of CEVE incorporation within the copolymeric chain. The monomer reactivity ratios were calculated along with structural parameters of the copolymers, such as the mean sequence lengths and the distribution of the monomer dyads. These structures may be further employed as scaffolds for the synthesis of graft copolymers. In particular, the pendant chlorine groups can be transformed to other useful groups, such as azides, hydroxyls, and bromines [20]. Therefore, they can be employed for click reactions (grafting onto procedure) and for Ring Opening (ROP) and Atom Transfer Radical Polymerization (ATRP) reactions (grafting from procedures), leading to various amphiphilic graft copolymers and smart materials offering temperature or pH response. These materials can be possible candidates for drug delivery studies.

## 2. Materials and Methods

### 2.1. Materials

To avoid the presence of oxygen and humidity, all procedures were carried out with high-vacuum techniques [34]. Prior to use, the NVP (≥97% FLUCA) and CEVE (99% MERCK) monomers were vacuum distilled after previously being dried over calcium hydride overnight. Azobisisobutyronitrile AIBN (98% ACROS) was purified by recrystallization twice from methanol and then vacuum dried. The Chain Transfer Agents (CTAs) [(O–ethylxanthyl)methyl]benzene (CTA-1) and O–ethyl S–(phthalimidymethyl) xanthate (CTA-2) were synthesized following literature procedures [35,36]. 1,4–Dioxane (FISHER) was purified through a basic alumina column, and N,N-dimethyl formamide DMF (MERCK) was vacuum distilled after drying over 4 Å molecular sieves overnight. The rest of the reagents and solvents were of commercial grade and were used as received (LiOH 98% Aldrich, CHCl_3_, 99% Aldrich and CDCl_3_, 99.8% Aldrich).

### 2.2. Copolymerization of NVP and CEVE via Radical RAFT Polymerization

The copolymerization reactions for the synthesis of the statistical copolymers of P(NVP–stat–CEVE) were performed in custom-made glass reactors under high vacuum (10^−6^ mmHg). Following loading with the polymerization mixture, the reactors were connected to a high-vacuum line, and the polymerization mixture was subjected to three freeze–vacuum–thaw pump cycles to eliminate the oxygen from within before being flamed-sealed and placed in a preheated oil bath.

To determine optimal polymerization conditions (regarding the polymerization temperature, the nature of the CTA, the monomer concentrations, the nature of the solvent, and the presence of additives in order to achieve the best molecular characteristics and the highest level of CEVE incorporation within the copolymer chain), several copolymerization reactions were conducted either in 1,4–dioxane, in bulk, or in DMF (monomer concentration from 4% up to 20% wt/v) with or without the addition of LiOH. In particular, three alternative monomer ratios of 100 mol, 300 mol, and 500 mol per 1 mol of the RAFT agent CTA-1 or CTA-2 and 0.2 mol of AIBN as the initiator were examined.

The copolymerization reactions took place at 60 °C and 90 °C and were terminated by removing the reactors from the oil bath and allowing the mixture to cool under cold water flow. The reactors were then opened, and the mixture was exposed to air.

The obtained polymers were precipitated in n–heptane and purified by repeated dissolutions in chloroform and re-precipitations in n–heptane to ensure that the unreacted monomer residues were quantitatively removed, followed by overnight vacuum drying to evaporate any residual solvent.

### 2.3. Reactivity Ratios of NVP and CEVE Using CTA-1 as RAFT Agent

A set of five copolymers of NVP and CEVE was prepared in bulk using CTA-1 at 90 °C to study the reactivity ratios of the monomers. Each copolymerization involved a different feed ratio (monomer molar ratios NVP/CEVE: 20/80, 40/60, 50/50, 60/40, and 80/20). In a typical process, 500 mol of the two monomers (e.g., 100 mol NVP and 400 mol CEVE for the sample 20–80) was added, together with 1 mol of CTA-1 and 0.2 mol AIBN in the reactor. The polymerization time depends on the copolymerization kinetics and the requirement to maintain low conversions, in order to achieve the highest degree of accuracy in the reactivity ratio measurements.

The final products were identified by the different molar feed ratios of the monomers; for instance, sample P(NVP–stat–CEVE)/CTA-1 40/60 corresponds to the copolymer synthesized utilizing CTA-1 as the RAFT agent and 40% NVP/60% CEVE as the molar feed composition. Size Exclusion Chromatography (SEC, Waters, Milford, MA, USA) and ^1^H–NMR spectroscopy (Bruker, Billerica, MA, USA) were used to monitor the copolymerization process. To estimate the monomer reactivity ratios, the experimental results were processed using the Fineman–Ross (F–R), inverted Fineman–Ross (inv. F–R), and Kelen–Tüdos (K–T) equations, as well as the COPOINT program.

### 2.4. Characterization Techniques

SEC experiments were conducted using a modular instrument composed of a Waters model 510 pump, a U6K sample injector, a 401 differential refractometer, and a set of 5 μm Styragel columns with a continuous porosity range from 500 to 10^6^ Å. The flow rate of the carrier solvent chloroform was equal to 1 mL/min. Nine polystyrene standards with molecular weights between 970 and 600,000 were used to calibrate the system. SEC was employed to determine the molecular characteristics of the copolymers, such as the weight-average molecular weight (M_w_) and the molecular weight distribution, Đ = M_w_/M_n_.

The composition of the copolymers was determined from their ^1^H–NMR spectra, which were recorded in chloroform–d at 298 K with a 400 MHz Bruker Avance Neo instrument.

Using a Q200 DSC model from TA Instruments (New Castle, DE, USA), the glass-transition temperatures were obtained by Differential Scanning Calorimetry (DSC). The samples were heated under nitrogen atmosphere at a rate of 10 °C/min from −100 °C to 200 °C. Results from the second heating were obtained in all cases.

Thermogravimetric analysis (TGA) was carried out on a Q50 TGA model from TA Instruments (New Castle, DE, USA) to examine the thermal stability and the kinetics of thermal decomposition of the copolymers employed. The copolymers were placed in a platinum pan and heated at heating rates of 3, 5, 7, 10, 15, and 20 °C/min under N_2_ flow at 60 mL/min, from ambient temperature to 800 °C.

## 3. Results and Discussion

### 3.1. Statistical Copolymers of NVP and CEVE via Radical RAFT Polymerization

The radical copolymerization of NVP and CEVE was carried out in a variety of feed molar ratios, solvents, and temperatures in the presence of two different CTAs in order to examine which one is best suited for the copolymerization reaction. Since these two monomers had never been copolymerized before, and since CEVE was considered to be capable of only cationic polymerization, a series of trial copolymerizations was performed to determine the optimal experimental conditions in order to achieve the most efficient incorporation of both monomers into the copolymeric structure (Figure 1).

The molecular characteristics of the trial copolymers were estimated by SEC using chloroform as the carrier solvent (Table 1). At 60 °C, employing a molar ratio of NVP and CEVE equal to 20/80, no polymer was obtained (Table 1, entry 1). The experiment was repeated, raising the temperature to 90 °C. After 48 h of reaction a copolymer was finally precipitated (Table 1, entry 2). The conversion was measured gravimetrically and was found to be equal to 16.5%. The copolymer’s composition was the exact opposite of the feed ratio, meaning that the incorporation of NVP was almost quantitative in the copolymeric structure, and that only 25% of the amount of CEVE was finally copolymerized. This observation was initially considered to be reasonable, since the radical polymerization of CEVE was accepted to be impossible. However, polymerization was possible, although with a significantly slower rate than that of NVP.

Tripling the concentration of both monomers and keeping the same monomer molar feed ratio equal to 20/80 seemed to have a minor effect regarding copolymer composition, leading to a small but noticeable increase in the CEVE incorporation into the copolymeric structure (Table 1, entry 3). However, a substantial increase was observed in the molecular weight of the copolymer, since it was found to increase twice as much as the previous sample.

When the same monomer molar ratio was employed in bulk, the results were slightly improved (Table 1, entry 6). In particular, the molecular weight increased, while the dispersity was almost the same as that measured when the copolymerization was conducted in 1,4-dioxane. In addition, both the conversion and the CEVE degree of incorporation increased sufficiently.

An additional increase in the quantities of the monomers up to 5 times compared to the sample described in Table 1, entry 1 was followed by an increase in the molecular weight of the copolymer, an even higher incorporation of CEVE units to the copolymeric structure, and higher conversion (Table 1, entry 4). In this case, the copolymerization was conducted in bulk. The results were improved compared to those presented in entries 3 and 6. The dispersity was only increased performing the copolymerization in bulk and employing higher amounts of monomers. Nevertheless, the absence of solvent and the copolymerization temperature at 90 °C seem to be the optimal experimental conditions for the copolymerization reaction. The increase in the quantities of monomers over that of the initiator and CTA leads to progressively increased molecular weights, as expected.

Experiments were also conducted with different monomer feed molar ratios. Instead of the NVP/CEVE molar ratio equal to 1/4, the ratio 1/1 was also employed, with the total monomer concentration over the CTA equal to 300/1 or 500/1 (Table 1, entries 5 and 7). A direct comparison of the results given at entries 3 and 5, having the same total monomer concentrations, reveals that the increase in the quantity of NVP leads to a very significant increase in the conversion and the molecular weight as well. It is obvious that NVP is a highly polymerizable monomer under these experimental conditions, and therefore it is reasonable to expect these changes. The composition in CEVE was slightly lower than that of entry 3. However, taking into account the lower feed ratio in CEVE, it can be concluded that the degree of incorporation of CEVE to the copolymeric structure was satisfactory. The only drawback of this procedure was the increased dispersity of the copolymer, which was attributed to the higher polymerizability of the NVP monomer and the increased conversion leading to several termination reactions causing the broadening of the molecular weight distribution. 

Upon increasing the total monomer concentration over the CTA to 500/1 again with equal molar ratios of NVP and CEVE (the ratio was 250/250, Table 1, entry 7), the conversion was also very high, as in the experiment described in entry 5. However, in the present case this result was achieved after 24 h of reaction. The molecular weight was further increased, as was expected considering the increased monomer quantities over the CTA. The major drawbacks of this copolymerization include the lower incorporation of the CEVE monomer units along the copolymer chains and the substantially increased dispersity. These results may be attributed to the employment of CTA-2 instead of CTA-1. Therefore, CTA-1 seems to be more appropriate to promote the copolymerization of NVP and CEVE, taking into account the following parameters: molecular weight, molecular weight distribution, conversion, and degree of incorporation of CEVE into the copolymeric structure. 

The radical polymerization of VEs was reported to be promoted in the presence of LiOH in aqueous solution [33]. In this study, the use of AIBN as the radical initiator eliminates the possibility of conducting the copolymerization reactions in water. Consequently, DMF was chosen as the polar solvent to test the performance of LiOH as an additive in the copolymerization of NVP and CEVE. Three experiments were carried out with the molar ratio [NVP]/[CEVE]/[CTA] = 60/240/1 using progressively increased quantities of LiOH (Table 1, entries 8, 9, and 10). Comparing these data with the results given in Table 1, entry 3, it can be concluded that the presence of LiOH does not offer a great benefit to the copolymerization reaction. In the presence of the LiOH, the calculated molecular weights were lower, the distribution was higher, the conversion was substantially lower, and the composition in CEVE much lower as well. Upon increasing the amount of the additive no considerable improvement was observed. Similar conclusions were also drawn using the molar ratio [NVP]/[CEVE]/[CTA] = 250/250/1 in the presence of LiOH (Table 1, entry 11). These results were compared with those presented in entry 7. The only improvement was the lower dispersity of the copolymer produced in the presence of the additive. However, this behavior is attributed to the fact that this copolymerization was conducted in solution, whereas that in entry 7 was conducted in bulk.

In any case, the use of LiOH as an additive during the copolymerization reaction between NVP and CEVE does not seem to offer any major advantage in organic solvents. This result can be attributed to the fact that in organic solvents there are no free Li cations. It was found that the presence of free Li^+^ is necessary to control the radical polymerization of vinyl ether, since the interactions of the Li cations with the oxygen group of the VE monomer stabilizes the growing radicals.

### 3.2. Monomer Reactivity Ratios and Statistical Analysis of the Copolymers

The Radical RAFT polymerization of NVP and CEVE was carried out in bulk at 90 °C with CTA-1 and AIBN as the polymerization initiator in the absence of any additive, taking into account the results previously reported. An extra set of five copolymers was synthesized for the determination of the monomers’ reactivity ratios, as described in the experimental section (Section 2.3). The polymerization times varied from 255 min when CEVE was the major component (NVP to CEVE molar ratio equal to 20/80) to just 10 min in the opposite case (NVP to CEVE molar ratio equal to 80/20).

The molecular characteristics of the copolymers were estimated by SEC and are provided in Table 2. The SEC traces are given for all samples in Figure 1. The molecular weight of the product 80/20 was significantly higher than that of the other samples, which is reasonable and expected, given that the copolymer is almost entirely composed of monomeric units of NVP (97.50% mol of NVP). The dispersity values range from 1.51 to 1.80, a relatively narrow range in spite of employing monomers of very different reactivities, including the VE, which was not considered susceptible to radical polymerization. The copolymerization time was arranged so that the conversion should be comparable for all polymers and relatively low, in order to apply the copolymerization equation and linear methods for the calculation of the monomer reactivity ratios.

The copolymer composition was calculated from the ^1^H–NMR spectra of the copolymers. A typical case is illustrated in Figure 2. The composition was calculated using the signals 10 and (1 + 7 + 14 + 15), which are assigned to the NVP and CEVE monomer units, as shown in Figure 2.

The NVP and CEVE reactivity ratios were determined using the Fineman–Ross (F–R) [37], inverted Fineman–Ross (inv. F–R) [37], and Kelen–Tüdos (K–T) [38] methods, along with the computer program COPOINT [39]. All the monomer reactivity ratios were calculated in accordance with the terminal model [40,41].

The F–R methodology states that the monomer reactivity ratios can be determined by the following equation:(1)$$G={\mathrm{Hr}}_{NVP}-r_{CEVE}$$
where G and H are
(2)$$G=\frac{X\left(Y-1\right)}{Y}$$
and
(3)$$H=\frac{X^{2}}{Y}$$
with
(4)$$X=\frac{M_{\mathrm{NVP}}}{M_{\mathrm{CEVE}}}$$
and
(5)$$Y=\frac{{\mathrm{dM}}_{\mathrm{NVP}}}{{\mathrm{dM}}_{\mathrm{CEVE}}}$$

M_NVP_ and M_CEVE_ are the monomer molar feed ratios and dM_NVP_ and dM_CEVE_ the copolymer compositions as calculated by ^1^H–NMR spectra.

The inv. F–R methodology is dependent on the following expression:(6)$$\frac{G}{H}=r_{NVP}-\frac{1}{H}r_{CEVE}$$

The intercept and slope of the graphs for both the G versus H values and the G/H versus 1/H plots offer the reactivity ratios r_NVP_ and r_CEVE_.

Alternatively, the reactivity ratios can be obtained using the K–T method, which can be summarized by the following equation:(7)$$\eta =\left(r_{NVP}+\frac{r_{CEVE}}{\alpha }\right)\xi -\frac{r_{CEVE}}{\alpha }$$

The η and ξ variables are functions of the G and H parameters, and their definitions are as follows:(8)$$\eta =\frac{G}{\alpha +H}$$
and
(9)$$\xi =\frac{H}{\alpha +H}$$

The α variable is a constant, which is equal to (H_max_H_min_)^1/2^, where H_max_ and H_min_ are the maximum and the minimum H values from the series of measurements, respectively. The intercept of the linear plot η versus ξ equals (−r_CEVE_/α), whereas the value of the function η at ξ = 1 equals r_NVP_. The K–T methodology is characterized by the fact that it assigns equal weight to all data points, yielding more realistic results than other methods. The F–R, inv-F–R, and K–T plots are given in the Appendix A.

Since the independent variable of the linear equations is not truly independent, and the variance of the dependent variable is not constant, all the employed linear methods have statistical constraints inherent to the applied linearization. On the contrary, the computer program COPOINT assesses the copolymerization parameters based on the monomer feed ratio, the copolymer composition data, and the copolymerization conversion as acquired from the copolymerization experiments. The program applies numeric integration techniques in their differential forms. The mathematical treatment can be theoretically used up to full monomer conversion, but it is advised that the conversion should not exceed 30%. By minimizing the sum of square differences between measured and calculated polymer compositions, the copolymerization parameters can be found.

The reactivity ratios of NVP and CEVE copolymerized in bulk at 90 °C via Radical RAFT polymerization using CTA-1 as the RAFT agent are provided in Table 3.

There are several reports regarding the synthesis of statistical copolymers based on NVP and methacrylates [2,3,4,5,6]. In all cases the NVP was the less activated of the two monomers. In this study, it is, to our knowledge, the first time where NVP was the more active of the two monomers in the copolymerization of NVP and CEVE. Since polymethacrylates are hydrophobic, like PCEVE, there was a higher rate of incorporation of the hydrophobic monomers in previous studies (as a result of higher activity). In contrast, in our study, macromolecular chains primarily composed of NVP hydrophilic monomer units were afforded.

The reactivity ratio of NVP not only significantly exceeded the unit but was even higher than 10 (r_NVP_ = 10.90), based on the COPOINT results, whereas the CEVE equivalent was much smaller (r_CEVE_ = 0.06). This indicates a definite preference of NVP homopolymerization at first, followed by a gradual conversion of the CEVE monomer into the copolymer. This gradual integration of CEVE into the macromolecular chain following the homopolymerization of the NVP units reveals that the copolymers can be considered gradient copolymers or pseudo-diblocks.

The conclusions above were also confirmed by calculating the structural properties of the copolymers. Using the equations suggested by Igarashi [42], it was possible to determine the dyad monomer sequences M_NVP_–M_NVP_, M_CEVE_–M_CEVE_, and M_NVP_–M_CEVE_:(10)$$X=\varphi_{NVP}-\frac{2\varphi_{NVP}\left(1-\varphi_{NVP}\right)}{1+{\left[{\left(2\varphi_{NVP}-1\right)}^{2}+4r_{NVP}r_{CEVE}\varphi_{NVP}\left(1-\varphi_{NVP}\right)\right]}^{\frac{1}{2}}}$$
(11)$$\;Y=\left(1-\varphi_{NVP}\right)-\frac{2\varphi_{NVP}\left(1-\varphi_{NVP}\right)}{1+{\left[{\left(2\varphi_{NVP}-1\right)}^{2}+4r_{NVP}r_{CEVE}\varphi_{NVP}\left(1-\varphi_{NVP}\right)\right]}^{\frac{1}{2}}}$$
(12)$$Z=\frac{4\varphi_{NVP}\left(1-\varphi_{NVP}\right)}{1+{\left[{\left(2\varphi_{NVP}-1\right)}^{2}+4r_{NVP}r_{CEVE}\varphi_{NVP}\left(1-\varphi_{NVP}\right)\right]}^{\frac{1}{2}}}$$
where X, Y, and Z are the mole fractions of the M_NVP_–M_NVP_, M_CEVE_–M_CEVE_, and M_NVP_–M_CEVE_ dyads, respectively, and φ_NVP_ is the NVP mole fraction in the copolymer. Additionally, the mean sequence lengths μ_NVP_ and μ_CEVE_ were calculated using the following equations:(13)$$\mu_{NVP}=1+r_{NVP}\frac{\left[NVP\right]}{\left[CEVE\right]}$$
(14)$$\mu_{CEVE}=1+r_{CEVE}\frac{\left[CEVE\right]}{\left[NVP\right]}$$

The r_NVP_ and r_CEVE_, for the calculations of the dyads and the mean sequence lengths, are those from COPOINT. The calculated data are summarized in Table 4 and Figure 3.

### 3.3. Glass Transition Temperatures of the Statistical Copolymers

The glass transition temperatures (T_g_) of the statistical copolymers P(NVP–stat–CEVE)/CTA-1 were determined by DSC. According to the literature, the T_g_ of PNVP is equal to 187 °C [10], whereas that of PCEVE is equal to −22 °C [20]. From the DSC measurements, there was just one glass transition temperature for each copolymer, which was found to be between the corresponding homopolymers’ extreme values. The T_g_ increased steadily as the amount of the NVP that integrated into the copolymer chain increased. The DSC plots are shown in the SM. This result indicates that there is no microphase separation, most likely due to the low molecular weights of the samples, facilitating the mixing process.

In general, the thermal properties of the copolymers are influenced by the chemical structure of their monomeric units, their composition, and their monomer sequence distribution. To anticipate the T_g_ value of any copolymer composition, several theoretical equations have been developed to describe how these factors potentially affect T_g_ values.

One example is the Gibbs–Di Marzio equation [43]:(15)$$T_{g}=m_{NVP}T_{g_{NVP}}+m_{CEVE}T_{g_{CEVE}}$$
where m_NVP_ and m_CEVE_ are the mole fractions of NVP and CEVE, respectively, in the copolymers, while Tg_NVP_ and Tg_CEVE_ are the glass transition temperatures of the corresponding homopolymers.

Another mathematical formula was proposed by Fox [44]:(16)$$\frac{1}{T_{g}}=\frac{w_{NVP}}{T_{g_{CEVE}}}+\frac{w_{CEVE}}{T_{g_{CEVE}}}$$
where w_NVP_ and w_CEVE_ are the weight fractions of NVP and CEVE, respectively, in the copolymers.

Both approaches depend only on thermodynamic and free value theories and take into account only the composition of the copolymers. On the other hand, Barton and Johnston’s approaches also include the monomer sequence distribution in addition to the impact of their compatibility on steric and energetic interactions.

The Barton equation indicates that [45]
(17)$$T_{g}=XT_{g_{NVP-NVP}}+YT_{g_{CEVE-CEVE}}+ZT_{g_{NVP-CEVE}}$$
where X, Y, and Z are the monomer dyad fractions (Equations (10)–(12), respectively).

In addition, Johnston reported the following equation [46]:(18)$$\frac{1}{Tg}=\frac{W_{NVP}P_{NVP-NVP}}{T_{g_{NVP-NVP}}}+\frac{W_{CEVE}P_{CEVE-CEVE}}{T_{g_{CEVE-CEVE}}}+\frac{W_{NVP}P_{NVP-CEVE}+W_{CEVE}P_{CEVE-NVP}}{T_{g_{NVP-CEVE}}}$$

It is assumed that NVP–NVP, CEVE–CEVE, and NVP–CEVE or CEVE–NVP dyads each have their own glass transition temperatures, T_gNVP–NVP_, T_gCEVE–CEVE_, and T_gNVP–CEVE_, respectively. It is reasonable to consider that the values of T_gNVP–NVP_ and T_gCEVE–CEVE_ correspond to the glass transition temperatures of the respective homopolymers, while T_gNVP–CEVE_ denotes the glass transition temperature of the alternating copolymer P(NVP–alt–CEVE). Regarding P_NVP–NVP_, P_CEVE–CEVE_, P_NVP–CEVE_, and P_CEVE–NVP_, those are the probabilities of having the corresponding dyad linkages. These probabilities can be determined using the following equations:(19)$$P_{\mathrm{NVP}-\mathrm{NVP}}=\frac{r_{\mathrm{NVP}}}{r_{\mathrm{NVP}}+\frac{\left[M_{\mathrm{CEVE}}\right]}{\left[M_{\mathrm{NVP}}\right]}}$$
(20)$$P_{\mathrm{NVP}-\mathrm{CEVE}}=\frac{\left[M_{\mathrm{CEVE}}\right]}{r_{\mathrm{NVP}}\left[M_{\mathrm{NVP}}\right]+\left[M_{\mathrm{CEVE}}\right]}$$
(21)$$P_{CEVE-NVP}=\frac{\left[M_{NVP}\right]}{r_{CEVE}\left[M_{CEVE}\right]+\left[M_{NVP}\right]}$$
(22)$$P_{CEVE-CEVE}=\frac{r_{CEVE}\left[M_{CEVE}\right]}{r_{CEVE}\left[M_{CEVE}\right]+\left[M_{NVP}\right]}$$

It is vital to know the glass transition temperature of the alternating copolymer in order to apply these methods; however, this value has not been reported in the literature. The T_gNVP–CEVE_ values were thus obtained using the linearized forms of the Johnston and Barton equations. The plots are given in Figure 4 and Figure 5. It is obvious that straight lines passing through the origin were obtained, indicating that these theoretical methodologies can accurately estimate the T_g_ values of statistical copolymers and therefore that the monomer sequence distribution is a significant factor influencing the T_g_ of the statistical copolymers. The T_gNVP–CEVE_ values calculated by the Johnston and Barton equations were 285.7 K and 263.9 K, respectively. Both the experimental T_g_ values and those predicted by all of the aforementioned approaches are given in Table 5.

After comparing the experimental results with the theoretically expected values from the various models, the Johnston equation was found to be the most suitable for the copolymers P(NVP–stat–CEVE), since the theoretically calculated values do not significantly exclude the experimentally acquired data in all ratios and the plot’s linearity is excellent (R^2^ = 0.99). This implies that the glass transition temperature is greatly influenced by the arrangement of the monomer units along the copolymer chain. Since the Johnston approach outperforms the Barton method for forecasting T_g_ values, use of the weight fraction recommendation is preferable for these copolymers.

### 3.4. Kinetics of the Thermal Decomposition of the Statistical Copolymers

The thermal stability and the kinetics of the thermal degradation of the statistical copolymers were investigated employing TGA and DTG measurements. According to the literature, the PNVP homopolymer exhibits a one-step thermal decomposition process at temperatures ranging from 416 °C to 453 °C [12]. Conversely, the PCEVE homopolymer has far more complicated thermal degradation behavior. DTG profiles in particular displayed a three-step degradation process [47]. The first phase is situated between 219 °C and 258 °C. The second and principal degradation process occurs between 291 °C and 329 °C, while the final and minor decomposition step is recorded around 434 °C to 453 °C. The data for PCEVE are given in the SM [47].

The first degradation peak of the statistical copolymers synthesized in this study is attributed to the CEVE monomer units. In fact, the contribution of the peak increases in direct proportion to the CEVE composition in the copolymers. In addition, the peak is visible even at extremely low amounts of CEVE, due to the pseudo-diblock structure, which was previously established from the examination of the reactivity ratios. Simultaneously, as the fraction of CEVE decreases, the peak temperature increases because the CEVE units are surrounded by thermally stable NVP units. As a result, the lower the percentage of CEVE is, the more thermally resistant its units become.

The second decomposition peak correlates directly to the primary degradation step of PNVP and the third degradation peak of PCEVE. This peak dominates the thermal degradation profile of the copolymers, since the NVP units prevail along the copolymeric chain for all copolymers. For the same reason, the peak temperatures do not vary considerably with composition. Furthermore, since this temperature is higher than that of the corresponding PNVP homopolymer, it can be concluded that the CEVE units boost the thermal stability of the NVP sequences.

The third and final peak is located at extremely high temperatures and has a wide temperature range (well above 600 °C in all samples and heating rates). Evidently, the residues of the thermal decomposition of NVP and CEVE produce highly thermally stable structures.

The DTG profiles for product 20/80 under various heating rates are given in Figure 6, whereas the corresponding profiles for all the statistical copolymers at the heating rate of 10 °C/min are in Figure 7. More data for the other statistical copolymers are provided in the SM.

The activation energies, E_a_, of the thermal decomposition procedure for the statistical copolymers were calculated using the well-established isoconversional Ozawa–Flynn–Wall (OFW) and Kissinger–Akahira–Sunose (KAS) methods. The complexity of the degradation mechanism renders the Kissinger method impractical for calculating the E_a_; thus, the other two methods were applied.

The conversion (α) and temperature (T) are used to express the reaction rate of the thermal decomposition reaction as follows:(23)$$\frac{d\alpha }{dt}=f\left(\alpha \right)k\left(T\right)$$
where (t) is time and f(α) is the differential conversion function.

The temperature dependence can be represented using an Arrhenius equation, which is
(24)$$k\left(T\right)=Ae^{-\frac{E_{\alpha }}{RT}}$$
where A is the pre-exponential factor ($\frac{1}{min}$), E_a_ is the activation energy ($\frac{KJ}{mol}$), and R is the gas constant (8.314 $\frac{J}{mol*K}$). If (23) is added to (24), it yields
(25)$$\frac{d\alpha }{dt}=Ae^{-\frac{E_{\alpha }}{RT}}f\left(\alpha \right)$$

Assuming a constant heating rate, as in this study, then
(26)$$\beta =\frac{dT}{dt}$$

Equation (25) is transformed to
(27)$$\frac{d\alpha }{dT}=\left(\frac{A}{\beta }\right)e^{-\frac{E_{\alpha }}{RT}}f\left(\alpha \right)$$

Rearranging Equation (27), the following equation is obtained:(28)$$\frac{d\alpha }{f\left(\alpha \right)}=\left(\frac{A}{\beta }\right)e^{-\frac{E_{\alpha }}{RT}}dT$$

Equation (28) is integrated to generate the following outcome:(29)$$g\left(\alpha \right)=\int_{0}^{\alpha }\frac{d\alpha }{f\left(\alpha \right)}=\frac{A}{\beta }\int_{T_{0}}^{T}e^{\frac{-E_{\alpha }}{RT}}dT=\frac{AE_{\alpha }}{\beta R}P\left(\chi \right)$$
where T_o_ and T are the reaction’s initiating and terminating temperatures, respectively, g(α) is the integral conversion, and χ equals ($\frac{Ea}{RT})$ [48,49,50,51,52,53].

A complete evaluation of the thermal degradation of a polymeric substance necessitates the determination of the activation energy, E_a_, and the pro-exponential factor A, alongside the mechanism or preferably the mathematical model of the thermal decomposition process. It is obvious that g(α) relies on the conversion mechanism and its mathematical model. A number of algebraic formulations have been developed for the functions of the most prevalent reaction mechanisms that are involved in solid-state reactions [54]. There is no analytical solution to the P(x) function. As a result, several approximations have been proposed, including the following:(30)$$P\left(\chi \right)=0.0048e^{\left(1.0516\chi \right)}$$
and
(31)$$P\left(\chi \right)=\frac{e^{-\chi }}{\chi^{2}}$$

Equation (30) is referred to as the Coates–Redfern and Equation (31) as the Doyle approximations. The OFW [55,56,57] and KAS [58] equations are produced by substituting Equations (30) and (31) to Equation (29):(32)$$(\mathrm{OFW}):\;\ln\beta =\ln\left[\frac{0.0048{\mathrm{AE}}_{\alpha }}{g\left(\alpha \right)R}\right]-1.0516\frac{E_{\alpha }}{\mathrm{RT}}$$
(33)$$(\mathrm{KAS}):\;\ln\frac{\beta }{T^{2}}=\ln\left[\frac{\mathrm{AR}}{g\left(\alpha \right)E_{\alpha }}\right]-\frac{E_{\alpha }}{\mathrm{RT}}$$

These approaches are “model free” methodologies and fall under the category of isoconversional approaches, since the conversion function f(α) is unaffected by variations in the heating rate, β, for all values of α. For this reason, displaying lnβ versus $\frac{1}{T}$ or ln($\frac{\beta }{T^{2}}$) versus $\frac{1}{T}$, respectively, should emerge in lines with slopes directly proportional to the activation energy. Moreover, a single-step degradation reaction can be inferred if the determined activation energy values do not significantly change with different values of α.

The OFW and KAS approaches entail measuring the temperatures that correspond to fixed values of α derived from experiments at various rates of heating, β. These methods can be used without knowledge of the reaction order of the decomposition process and are highly helpful for the kinetic interpretation of thermogravimetric data collected from complicated processes, such as the thermal degradation of polymers. The OFW method relies on the Coates and Redfern approximation (which leads to Equation (32)) [59], whereas the KAS method depends on the more accurate Doyle approximation (which leads to Equation (33)) [60]. In light of this, the latter approach is expected to provide greater precision in determining the activation energy of the thermal degradation process.

The activation energy values obtained by OFW and KAS methodologies are displayed in Table 6, whereas characteristic OFW and KAS plots for the sample 80/20 are given in Figure 8 and Figure 9. More plots are given in the SM.

As the values from OFW and KAS are generally comparable, it can be concluded that these methods are highly accurate at determining the activation energy at different stages of thermal deterioration.

The E_a_ values were lower at low heating rates because these correspond to the sequences of CEVE monomeric units, which are thermally less stable than the corresponding sequences of NVP. The low E_a_ values in the sample 20/80 with the highest percentage of CEVE correspond to weight loss corresponding to α = 0.1 and α = 0.2 values. In contrast, the E_a_ values are remarkably consistent during all stages of degradation in the copolymer 80/20 with the lowest proportion of CEVE.

Although less than that observed in the PCEVE homopolymer, the variation in E_a_ values continues to exceed that reported in the PNVP homopolymer. Because of the prolonged sequences of the structural units of NVP in the copolymers, the individual E_a_ values are clearly closer to the corresponding values of the homopolymer PNVP.

## 4. Conclusions

The statistical radical copolymerization of CEVE with NVP was accomplished via the RAFT polymerization technique without employing additives. The SEC analysis showed that the synthesized copolymers exhibited relatively high molecular weights with monomodal and symmetrical peaks having moderate values of dispersities. Sufficient incorporation of CEVE monomer units along the copolymeric chain was achieved (up to 34% by NM measurements). Amphiphilic statistical copolymers were produced, with the hydrophilic nature of the NVP monomer units predominating in all of them.

The monomer reactivity ratios were calculated using the F–R, inv. F–R, and K–T methodologies, as well as the computer program COPOINT. The results revealed that the copolymers can be considered as gradient copolymers or pseudo-diblocks. These results were further confirmed by measuring the dyad monomer fractions and the mean sequence lengths of the monomers.

The thermal decomposition studies of the statistical copolymers exhibited behavior which resembles that of the PNVP homopolymer, which is fully justified by the high incorporation of NVP. Therefore, a major decomposition peak is observed in DTG measurements for all samples. Nevertheless, even in the samples with the least composition of CEVE, another degradation peak at lower temperatures is still noticeable and is attributed to the sequences of CEVE. In fact, the presence of the monomer units of CEVE appears to shield those of NVP by increasing their thermal stability. The activation energies of the thermal decomposition of the statistical copolymers were measured using the isoconversional methods of Ozawa–Flynn–Wall (OFW) and Kissinger–Akahira–Sunose (KAS). The higher the incorporation of NVP monomer units was the less important the variation in the activation energy values of the copolymers was with the conversion of thermal degradation.

Finally, for the theoretical calculation of the glass transition temperature, several models were applied and compared with the experimental results. Among them, the Johnston method was in better agreement with the experimental data, showing that the Tg values of the copolymers are not only influenced by the composition of the copolymers but by the sequence of the monomer units along the copolymeric chain as well.

## Data Availability

The data are available to any request.

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
