# Peer review of "Statistical Copolymers of N–Vinylpyrrolidone and 2–Chloroethyl Vinyl Ether via Radical RAFT Polymerization: Monomer Reactivity Ratios, Thermal Properties, and Kinetics of Thermal Decomposition of the Statistical Copolymers"

_polymers, 2023, doi:10.3390/polym15081970_

Round 1
Reviewer 1 Report
In the submitted manuscript by Pitsikalis and Plachouras, statistical copolymerization of N-vinylpyrrolidone (NVP) and 2-chloroethyl vinyl ether (CEVE) was studied in bulk via radical reversible activation-fragmentation chain-transfer (RAFT) polymerization for the first time. In general, CEVE is used as a monomer in cationic polymerization, and is considered difficult to polymerize under radical conditions. In this manuscript, however, the authors are able to realize radical copolymerization of NVP and CEVE, although the incorporation ratio of CEVE is lower than the starting molar ratio. The molecular weight of the produced copolymer can be slightly increased by enhancing total monomer concentration but significantly increased when a higher NVP molar ratio is used. In addition, two RAFT chain transfer agents are used, but control on molecular weight and dispersity could not be achieved. Moreover, the reactivity ratios of NVP and CEVE are determined by a series of copolymerization started with different monomer molar ratios. Based on these experiments, NVP is much more reactive than CEVE (rNVP = 10.90 vs rCEVE = 0.06). The thermal properties of copolymers are further measured by differential scanning calorimetry (DSC). Only one glass transition is observed for each copolymer sample, and the Tg increases with higher NVP incorporation ratio. The kinetics of the thermal degradation are also studied by thermogravimetric analysis (TGA) and differential thermogravimetry (DTG). Overall, the authors have demonstrated the radical copolymerization of NVP and CEVE, calculated the reactivity ratios, as well as characterized the thermal properties of the copolymers. This work could be suitable for Polymers, after the following major issues are addressed.
• I don’t think either RAFT agents show a good control on the polymerization, because the dispersities are high and the obtained molecular weights are much lower than the theoretic molecular weights (considering the monomer/CTA ratio and monomer conversion). I wonder if a control experiment of free radical copolymerization can be performed and compared with the RAFT results.
• I am curious to see how a dithiocarbamate CTA behaves in this copolymerization. This type of CTA is also commonly used in less activated monomers.
• In the procedure of copolymerization in solution, either solvent volume or monomer concentration needs to be added.
• Line 59-60 – this sentence is misleading. What “has been reported only for hydroxy–functional VEs in aqueous media”? Does this refers to radical polymerization of CEVE, or beta-scission?
• “Finemann” is misspelled. It should be “Fineman”.
• Equations 5 and 11 – “Y” should be used instead of upsilon.
• Equations 7, 23, 25, 27, 29, 32, 33, also in line 485 – “a” should be alpha as the conversion.
• Line 272 – “Hmin are” should be “Hmin are”.
• Line 479-481 – This paragraph is repeated.
• Line 496 – I don’t think this is a controlled polymerization.
Author Response
Reviewer #1
In the submitted manuscript by Pitsikalis and Plachouras, statistical copolymerization of N-vinylpyrrolidone (NVP) and 2-chloroethyl vinyl ether (CEVE) was studied in bulk via radical reversible activation-fragmentation chain-transfer (RAFT) polymerization for the first time. In general, CEVE is used as a monomer in cationic polymerization, and is considered difficult to polymerize under radical conditions. In this manuscript, however, the authors are able to realize radical copolymerization of NVP and CEVE, although the incorporation ratio of CEVE is lower than the starting molar ratio. The molecular weight of the produced copolymer can be slightly increased by enhancing total monomer concentration but significantly increased when a higher NVP molar ratio is used. In addition, two RAFT chain transfer agents are used, but control on molecular weight and dispersity could not be achieved. Moreover, the reactivity ratios of NVP and CEVE are determined by a series of copolymerization started with different monomer molar ratios. Based on these experiments, NVP is much more reactive than CEVE (rNVP = 10.90 vs rCEVE = 0.06). The thermal properties of copolymers are further measured by differential scanning calorimetry (DSC). Only one glass transition is observed for each copolymer sample, and the Tg increases with higher NVP incorporation ratio. The kinetics of the thermal degradation are also studied by thermogravimetric analysis (TGA) and differential thermogravimetry (DTG). Overall, the authors have demonstrated the radical copolymerization of NVP and CEVE, calculated the reactivity ratios, as well as characterized the thermal properties of the copolymers. This work could be suitable for Polymers, after the following major issues are addressed.
Thank you very much for the generally positive comments to our manuscript. We recognize that some misleading points have to be clarified and certain mistakes have to be corrected.
- I don’t think either RAFT agents show a good control on the polymerization, because the dispersities are high and the obtained molecular weights are much lower than the theoretic molecular weights (considering the monomer/CTA ratio and monomer conversion). I wonder if a control experiment of free radical copolymerization can be performed and compared with the RAFT results.
We agree with the reviewer that the RAFT polymerization of CEVE is not controlled and due to this behavior, the copolymerization reaction with NVP presents many difficulties leading to poor control over the molecular weights, the dispersity and the composition. This is clearly stated in the text. However, the RAFT methodology provides certain advantages compared to conventional radical polymerization. In the latter case the copolymerization results in products with much higher dispersity values (Ð values much higher than 2.0), lower molecular weights and much lower levels of CEVE incorporation into the copolymerization chain. In some cases, insoluble products were obtained. This result may be attributed to the existence of transfer reaction with the pendant chlorine groups of the CEVE monomer units. Definitely, the conventional radical polymerization fails to provide the desired structures.
- I am curious to see how a dithiocarbamate CTA behaves in this copolymerization. This type of CTA is also commonly used in less activated monomers.
We have not tried to use dithiocarbamates as CTAs for this type of copolymerization reactions. CTAs 1 and 2, employed in this work, provided relatively good control for the polymerization of NVP and that is they why were tested also in this study. We are in progress to study the statistical and block copolymerization reaction of NVP with other vinyl ethers as well. During this effort we plan to use other CTAs as well. The proposal of the reviewer seems a reasonable expansion of our work and we are thankful for this.
- In the procedure of copolymerization in solution, either solvent volume or monomer concentration needs to be added.
The copolymerization reactions, which provided the samples for the measurements of the reactivity ratios and for the study of the thermal properties were prepared in bulk. However, the reviewer is right for the test experiments. In certain cases, we employed the copolymerization reactions in dioxane or DMF solutions. The information concerning the monomer concentrations were added at the experimental section of the manuscript.
- Line 59-60 – this sentence is misleading. What “has been reported only for hydroxy–functional VEs in aqueous media”? Does this refers to radical polymerization of CEVE, or beta-scission?
The reviewer is right. The sentence is misleading. It was clarified in the revised text. We do not have any specific data for the conventional radical polymerization for CEVE. The typical behavior for all vinyl ethers is that they cannot stabilize the radicals and the system is susceptible to β-scission reactions. Only in the case of the hydroxy functionalized vinyl ether monomers the rather well-controlled conventional radical polymerization has been reported (references 31, 32).
- “Finemann” is misspelled. It should be “Fineman”.
We are sorry for the orthographical mistake. It was corrected in the revised text.
- Equations 5 and 11 – “Y” should be used instead of upsilon.
We are sorry for the mistake. Actually, it is not a mistake. The problem is that when writing the manuscript, we switched from the English keyboard to the corresponding Greek one. However, it is reasonable that the reviewer does not have the ability to recognize the Greek letters. Therefore, we used the corresponding symbols for these letters and hope that now there will be no problem.
- Equations 7, 23, 25, 27, 29, 32, 33, also in line 485 – “a” should be alpha as the conversion.
We are sorry for the mistake. Actually, it is not a mistake. The problem is that when writing the manuscript, we switched from the English keyboard to the corresponding Greek one. However, it is reasonable that the reviewer does not have the ability to recognize the Greek letters. Therefore, we used the corresponding symbols for these letters and hope that now there will be no problem.
- Line 272 – “Hmin are” should be “Hminare”.
We are sorry for the orthographical mistake. It was corrected in the revised text.
- Line 479-481 – This paragraph is repeated.
We are sorry for the mistake. The paragraph was deleted in the revised text.
- Line 496 – I don’t think this is a controlled polymerization.
The reviewer is right. It is not a controlled system. Therefore, the word “controlled” was deleted in the revised text.
Reviewer 2 Report
Manuscript polymers-2326478
The manuscript reports radical RAFT polymerization of NVP and CEVE and is focused on its optimization and the properties of the resulting copolymers. In general, the manuscript fits within the scope of Polymers. There are however some minor issues that should be addressed before the manuscript is reconsidered. Please refer to the comments below.
GENERAL COMMENTS
-slight editing of English language is necessary; it will not be comented
-please ensure that the capitalization of words throughout the manuscript is correct, e.g. capital letters in the names of techniques
-please shorten some of the keywords; the Authors can consider using commonly used abbreviations in this section
-while the Authors explained in the introduction why both polymers, PNVP and PCEVE, were considered and their applications, the explanation as to why the resulting copolymers would be significant is missing
-the email address should normally be sufficient, please remove the phone number
-it seems that the resulting products are “block copolymers” rather than statistical ones; could the Authors argument more their choice for the classification of their products? additionally, could the Authors explain more mentioning “pseudo-diblocks” or “gradient copolymers”?
-word “sample” seems to be overused throughout the manuscript and is confusing; consider replacing this word by its meaning in each case: e.g. product, monomer, copolymer, etc.
TITLE
-should be rewritten; in the second part of the title, are the terms addressing the properties of the resulting polymers or the monomers?
ABSTRACT
-please consider abbreviating the resulting polymer and state it in the abstract rather than addressing it as “Statistical Copolymers of NVP and CEVE”; this will allow the Reader to focus more on the content of the manuscript (expression “P(NVP-stat-CEVE)” is used in the caption of Fig1)
RESULTS AND DISCUSSION
-since the both polymers had not been copolymerized before and a new compound has been obtained, a typical characterization of the compound should be provided, or at least a full NMR spectrum to prove its structure and a mass spectrum to show their molecular structure, should be provided (e.g. in an SI part)
-what is the structure of the copolymer? please add a structural formula(s)
-line 176: the Reviewer suggests to change “the best” to “optimal”
-the whole discussion on p.5 should be rewritten; in the current form it is very general and difficult to follow; for example, in line 194, instead of “compared to entry 5” the Authors should quantify the high conversion rate in percentage
-lines 220 to 223: this part should be rewritten; also the reason why LiOH was used as an additive should be stated as well
-equations 1 to 9: all symbols used in the equations should be explained
-lines 252 to 255: this part should be moved to the experimental part; in its place, a short text explanation to the F-R methodology should be placed, which is going to make this fragment more interesting and easier to follow for a broader Readership
-equations 10 to 12: all new symbols should be explained
-in numerous cases, expressions from the caption of the figures or tables are not used or explained in the text, e.g. “Dyad sequence distribution”; both, the introduction and the experimental part should provide the Reader with all the necessary information to follow and understand the scope and content of the manuscript; both parts should be improved and/or reorganized
-DSC curves should be presented in an overlap format; due to one polymer being crystalline and the other one amorphous, it should be shown how their copolymerization affects the glass transition temperature of the resulting product; it will provide the visualization to a longer fragment of text, spanning over ca. 3 pages of manuscript
-the fonts size in both graphs in figures 4 and 5 is too small; the numerical values should have a period instead of a comma and the axes legends should include units next to them in brackets; please consider making the text uniform to give your results more relevance
-lines 391-392: this sentence can be removed
-please ensure that the figures in the SI part (the Reviewer recommend to change the abbreviation to “SI”, Supporting Information) are of a comparable quality to those in the manuscript file; please remove the excess of digits after comma, and change the comma style to period
-a scheme depicting the mechanism of the copolymerization should be placed in this section
CONCLUSIONS
-this part should be rewritten as a whole to fit the content of the manuscript
Author Response
Reviewer #2
The manuscript reports radical RAFT polymerization of NVP and CEVE and is focused on its optimization and the properties of the resulting copolymers. In general, the manuscript fits within the scope of Polymers. There are however some minor issues that should be addressed before the manuscript is reconsidered. Please refer to the comments below.
Thank you very much for the generally positive comments to our manuscript. We recognize that some misleading points have to be clarified and certain mistakes have to be corrected.
GENERAL COMMENTS
-slight editing of English language is necessary; it will not be commented
An English-speaking person was responsible for the editing of the text.
-please ensure that the capitalization of words throughout the manuscript is correct, e.g. capital letters in the names of techniques
We took into consideration the statement of the reviewer.
-please shorten some of the keywords; the Authors can consider using commonly used abbreviations in this section
We took into consideration the statement of the reviewer and we revised the keywords.
-while the Authors explained in the introduction why both polymers, PNVP and PCEVE, were considered and their applications, the explanation as to why the resulting copolymers would be significant is missing
The explanation was given at the end of the Introduction of the manuscript. However, this was not clear enough. The pendant chlorine groups can be transformed to other useful functional groups, e.g. azides, hydroxyls and bromides. They can be further used for click reactions (grafting onto procedure) and for ROP or ATRP reactions (grafting from procedures) leading to various amphiphilic graft copolymers and smart materials (temperature and pH-responsive systems). We have already started working towards this direction with very positive results. The appropriate revision was made in the text.
-the email address should normally be sufficient, please remove the phone number
In other papers the telephone number was required. Since the reviewer has a different opinion, we deleted the telephone number.
-it seems that the resulting products are “block copolymers” rather than statistical ones; could the Authors argument more their choice for the classification of their products? additionally, could the Authors explain more mentioning “pseudo-diblocks” or “gradient copolymers”?
The resulting products are not pure diblock copolymers. It is common to call these materials as either pseudo-diblock or gradient copolymers. The exact structure depends on the difference of the reactivity ratios of the monomers. When the copolymerization is initiated NVP monomer units are inserted exclusively in the polymer chain. However, as the quantity of the NVP monomer is depleting progressively CEVE units start to enter the copolymeric structure. Finally, after the consumption of the NVP monomer only CEVE units are incorporated. Therefore, the final structure is constructed by an almost pure block of PNVP (at the beginning) an almost pure block of PCEVE (at the end of the copolymer) and an intermediate block with both monomer present.
-word “sample” seems to be overused throughout the manuscript and is confusing; consider replacing this word by its meaning in each case: e.g. product, monomer, copolymer, etc.
We took into consideration the suggestion of the reviewer and made the appropriate changes in the text.
TITLE
-should be rewritten; in the second part of the title, are the terms addressing the properties of the resulting polymers or the monomers?
Obviously the second part of the title refers mainly to the copolymers. The title was changed to the following: “Statistical Copolymers of N–Vinylpyrrolidone and 2–Chloroethyl Vinyl Ether via Radical RAFT Polymerization: Monomer Reactivity Ratios, Thermal Properties and Kinetics of Thermal Decomposition of the Statistical Copolymers”
ABSTRACT
-please consider abbreviating the resulting polymer and state it in the abstract rather than addressing it as “Statistical Copolymers of NVP and CEVE”; this will allow the Reader to focus more on the content of the manuscript (expression “P(NVP-stat-CEVE)” is used in the caption of Fig1)
The same abbreviation for the statistical copolymers was used throughout the text.
RESULTS AND DISCUSSION
-since the both polymers had not been copolymerized before and a new compound has been obtained, a typical characterization of the compound should be provided, or at least a full NMR spectrum to prove its structure and a mass spectrum to show their molecular structure, should be provided (e.g. in an SI part)
A characteristic NMR spectrum of one of the copolymers is already included in the manuscript with the assignment of all the protons.
-what is the structure of the copolymer? please add a structural formula(s)
The structure is given in Figure 2, as insert of the NMR spectrum. However, the reviewer is right. It is more appropriate to give the structure in the main text of the manuscript. Therefore, Scheme 1 was revised appropriately.
-line 176: the Reviewer suggests to change “the best” to “optimal”
We made the suggested change.
-the whole discussion on p.5 should be rewritten; in the current form it is very general and difficult to follow; for example, in line 194, instead of “compared to entry 5” the Authors should quantify the high conversion rate in percentage
The text was rewritten to clarify the conducted experiments.
-lines 220 to 223: this part should be rewritten; also the reason why LiOH was used as an additive should be stated as well
The paragraph was revised to give a clearer message. In aqueous solutions Li+ are formed which interact with the oxygen atom of the vinyl ether monomer. These interactions promote the relative control over the radical polymerization of vinyl ethers (More details can be found in Ref. 33).
-equations 1 to 9: all symbols used in the equations should be explained
All the symbols were explained in the text. The problem is that when writing the manuscript, we switched from the English keyboard to the corresponding Greek one. However, it is reasonable that the reviewer does not have the ability to recognize the Greek letters. Therefore, we used the corresponding symbols for these letters and hope that now there will be no problem.
-lines 252 to 255: this part should be moved to the experimental part; in its place, a short text explanation to the F-R methodology should be placed, which is going to make this fragment more interesting and easier to follow for a broader Readership
We believe that this text is necessary at this point of the manuscript, since at the following paragraphs we analyze these methodologies, in order to help the broader readership of this work.
-equations 10 to 12: all new symbols should be explained
All the symbols were explained in the text. The problem is that when writing the manuscript, we switched from the English keyboard to the corresponding Greek one. However, it is reasonable that the reviewer does not have the ability to recognize the Greek letters. Therefore, we used the corresponding symbols for these letters and hope that now there will be no problem.
-in numerous cases, expressions from the caption of the figures or tables are not used or explained in the text, e.g. “Dyad sequence distribution”; both, the introduction and the experimental part should provide the Reader with all the necessary information to follow and understand the scope and content of the manuscript; both parts should be improved and/or reorganized.
We tried to reorganize these expressions throughout the text.
-DSC curves should be presented in an overlap format; due to one polymer being crystalline and the other one amorphous, it should be shown how their copolymerization affects the glass transition temperature of the resulting product; it will provide the visualization to a longer fragment of text, spanning over ca. 3 pages of manuscript
Both components of the statistical copolymers are amorphous. Neither block is crystalline. All the details are given in the main text of the manuscript. The DSC curves are provided at the Supporting Information section. In the main text we included only the results in Table 5. The relative discussion (pages 10-12) refers to the development of several theoretical models that can be adopted to predict the Tg value of any copolymer taking into account the Tg values of the respective homopolymers, the copolymer composition and the distributions of the monomer dyads along the copolymeric chain.
-the fonts size in both graphs in figures 4 and 5 is too small; the numerical values should have a period instead of a comma and the axes legends should include units next to them in brackets; please consider making the text uniform to give your results more relevance
The proposed changes have been made in the revised text.
-lines 391-392: this sentence can be removed
The sentence was removed as proposed by the reviewer.
-please ensure that the figures in the SI part (the Reviewer recommend to change the abbreviation to “SI”, Supporting Information) are of a comparable quality to those in the manuscript file; please remove the excess of digits after comma, and change the comma style to period
The proposed changes have been made in the revised manuscript.
-a scheme depicting the mechanism of the copolymerization should be placed in this section
The copolymerization mechanism is typical RAFT polymerization mechanism, which is available in many Polymer Chemistry textbooks. What is novel here is the application to the certain couple of monomers.
CONCLUSIONS
-this part should be rewritten as a whole to fit the content of the manuscript
The specific section was revised accordingly.
Round 2
Reviewer 1 Report
I'm glad to see that the authors have addressed my concerns and provided corresponding explanations on in their latest manuscript revision. Based on these revisions, I believe that the manuscript is now suitable for publication.
Author Response
We are thankful to the Reviewer for his very positive comments.
Reviewer 2 Report
Manuscript polymers-2326478 v2
This is the revised version of the manuscript reporting the radical RAFT polymerization of NVP and CEVE and the properties of the resulting statistical copolymers depending on the conditions of the polymerization conditions. While the manuscript did improve, not all remarks of the Reviewers have been fully addressed and there are some remaining comments to be addressed. Please refer to the remarks below.
GENERAL COMMENTS
-some changes seem to have not been marked in the manuscript
-not all comments of the Reviewers have been fully addressed
ABSTRACT
line 15: please remove “efficiently”
INTRODUCTION
line 39: please add a reference to the statement about the exclusivity of RAFT to obtain PNVP
-this part lacks an introduction to the research at the end; please add 1 or 2 sentences introducing your work and guiding the Reader through your results.
MATERIAL AND METHODS
line 91: please specify the remaining reagents and their source
line 96: “employing high vacuum–techniques.” please change to “under high vacuum” and specify the pressure level
line 100: please specify “optimal polymerization conditions”; what was the criterion the Authors were basing their conclusions on?
line 101: please add a sentence or a fragment explaining the difference between the following reaction conditions: “1,4–Dioxane, in bulk and in DMF”; please specify “the bulk” conditions and why were these particular (solvent) conditions chosen
line 144: “under N2 flow”
RESULTS AND DISCUSSION
line 166 and 167: two instances of “was considered” in one sentence
line 174: please specify which monomer
line 178: please specify the “slight improvement”
line 183: please consider not using expressions like “entry 1” in the text (except in brackets while referring to the table); a nice example of abbreviating the copolymers is in line 243 and abbreviated as “product 80/20” (please be consistent with the order of NVP and CEVE); an additional sentence in the experimental stating that copolymers would be referred in this manner, would clarify the discussion later on
-the Authors should be consistent with the use of the “entry”, as it seems to have different meaning throughout the manuscript; the Reviewer suggests not to use this term in the text in other format as (Table X., entry Y); an example for such a sentence: lines 192-194. The flow of the discussion would be thus improved and it would give more significance to the unique results obtained by the Authors; some examples are listed below:
-“as in the experiment described in entry 5”: refers to experiment setup
-“The composition in CEVE was slightly lower than that of entry 3” seems to refer to a copolymer
-“Direct comparison of the results given at entries 3 and 5, having the same total monomer concentrations,(…)
-“These results were compared with those presented in entry 7.”
-please be consistent how you refer to the molar ration throughout the text; at the moment, both xx:yy and y/x phrases are used, which seems to be confusing
lines 232: the 2 newly inserted sentences seem to be not related; please consider linking them by e.g., “On the other hand,…”
-Caption of Fig. 6: unnecessary capitalization of word “Methods”; please add an explanation of “A” in the caption
-“Ea”: “a” should be in the subscript
CONCLUSIONS
lines 527-528: “(rNVP=10.90 rCEVE=0.06 via the 528 COPOINT program)” can be removed, as it not fits the conclusion style
Author Response
We are thankful to Reviewer #2 for the detailed examination of our manuscript, his suggestions and criticisms. All the new changes to the second revision are given in red.
Reviewer #2
This is the revised version of the manuscript reporting the radical RAFT polymerization of NVP and CEVE and the properties of the resulting statistical copolymers depending on the conditions of the polymerization conditions. While the manuscript did improve, not all remarks of the Reviewers have been fully addressed and there are some remaining comments to be addressed. Please refer to the remarks below.
GENERAL COMMENTS
-some changes seem to have not been marked in the manuscript
-not all comments of the Reviewers have been fully addressed
ABSTRACT
line 15: please remove “efficiently”
The word was removed.
INTRODUCTION
line 39: please add a reference to the statement about the exclusivity of RAFT to obtain PNVP
We stated in the manuscript that NVP can be exclusively polymerized via radical polymerization. Among the various controlled radical polymerization techniques RAFT offers the best advantages. We added a reference about the first statement as was suggested by the Reviewer.
-this part lacks an introduction to the research at the end; please add 1 or 2 sentences introducing your work and guiding the Reader through your results.
A few sentences were added at the end of the Introduction section.
MATERIAL AND METHODS
line 91: please specify the remaining reagents and their source
We added the remaining reagents and their sources.
line 96: “employing high vacuum–techniques.” please change to “under high vacuum” and specify the pressure level.
The proposed change was made in the text.
line 100: please specify “optimal polymerization conditions”; what was the criterion the Authors were basing their conclusions on?
This point was further clarified in the revised text.
line 101: please add a sentence or a fragment explaining the difference between the following reaction conditions: “1,4–Dioxane, in bulk and in DMF”; please specify “the bulk” conditions and why were these particular (solvent) conditions chosen.
This point was further clarified in the revised text.
line 144: “under N2 flow”
We made the suggested correction.
RESULTS AND DISCUSSION
line 166 and 167: two instances of “was considered” in one sentence
We corrected the sentence.
line 174: please specify which monomer
We refer to both monomers. This point was clarified.
line 178: please specify the “slight improvement”
This point is already analyzed in the next sentences of the same paragraph. The comparison of the two samples (Table 1, entries 3 and 6) refers to their molecular weight, dispersity values, copolymerization conversion and level of CEVE incorporation within the copolymerization chain.
line 183: please consider not using expressions like “entry 1” in the text (except in brackets while referring to the table); a nice example of abbreviating the copolymers is in line 243 and abbreviated as “product 80/20” (please be consistent with the order of NVP and CEVE); an additional sentence in the experimental stating that copolymers would be referred in this manner, would clarify the discussion later on
-the Authors should be consistent with the use of the “entry”, as it seems to have different meaning throughout the manuscript; the Reviewer suggests not to use this term in the text in other format as (Table X., entry Y); an example for such a sentence: lines 192-194. The flow of the discussion would be thus improved and it would give more significance to the unique results obtained by the Authors; some examples are listed below:
-“as in the experiment described in entry 5”: refers to experiment setup
-“The composition in CEVE was slightly lower than that of entry 3” seems to refer to a copolymer
-“Direct comparison of the results given at entries 3 and 5, having the same total monomer concentrations,(…)
-“These results were compared with those presented in entry 7.”
It was clearly stated in the manuscript that before performing the copolymerization reactions for the determination of the reactivity ratios several trial experiments were conducted to find the optimal experimental conditions for the desired copolymerizations. These experiments were included in Table 1. Only for these samples, we used the entry number of the Table 1 to differentiate the samples and compare the results. For the reactivity ratio studies a different set of samples was synthesized taking into account the experimental data of Table 1. These final samples were named by the various monomer feed ratios. Always NVP was reported first.
-please be consistent how you refer to the molar ration throughout the text; at the moment, both xx:yy and y/x phrases are used, which seems to be confusing
We kept a consistent way of referring the samples.
lines 232: the 2 newly inserted sentences seem to be not related; please consider linking them by e.g., “On the other hand,…”
We made a suitable addition.
-Caption of Fig. 6: unnecessary capitalization of word “Methods”; please add an explanation of “A” in the caption
The Reviewer refers to Table 6 and not to Fig. 6. The appropriate changes have been made.
-“Ea”: “a” should be in the subscript
This was corrected throughout the text.
CONCLUSIONS
lines 527-528: “(rNVP=10.90 rCEVE=0.06 via the 528 COPOINT program)” can be removed, as it not fits the conclusion style
The part of the sentence was removed.